# Optimizing the microscopic agglutination test (MAT) panel for the diagnosis of Leptospirosis in a low resource, hyper-endemic setting with varied microgeographic variation in reactivity

**Dinesha Jayasundara**[1,2], **Chandika Gamage**[3], **Indika Senavirathna**[1,4], **Janith Warnasekara**[5], **Michael A. Matthias**[6], **Joseph M. Vinetz**[6‡]*, **Suneth Agampodi**[1,6‡]*

1 Leptospirosis Research Laboratory, Department of Community Medicine, Faculty of Medicine and Allied Sciences, Rajarata University of Sri Lanka, Saliyapura, Sri Lanka, 2 Department of Microbiology, Faculty of Medicine and Allied Sciences, Rajarata University of Sri Lanka, Saliyapura, Sri Lanka, 3 Department of Microbiology, Faculty of Medicine, University of Peradeniya, Peradeniya, Sri Lanka, 4 Department of Biochemistry, Faculty of Medicine and Allied Sciences, Rajarata University of Sri Lanka, Saliyapura, Sri Lanka, 5 Department of Community Medicine, Faculty of Medicine and Allied Sciences, Rajarata University of Sri Lanka, Saliyapura, Sri Lanka, 6 School of Medicine, Yale University, New Haven, Connecticut, United States of America

‡ These authors are joint senior authors on this work.
* joseph.vinetz@yale.edu (JV); suneth.agampodi@yale.edu (SA)

## Abstract

The microscopic agglutination test (MAT) is the standard serological reference test for the diagnosis of leptospirosis, despite being a technically demanding and laborious procedure. The use of a locally optimised MAT panel is considered essential for proper performance and interpretation of results. This paper describes the procedure of selecting such an optimised panel for Sri Lanka, a country hyper-endemic for leptospirosis. MAT was performed using 24 strains on 1132 serum samples collected from patients presenting with acute undifferentiated fever. Of 24 strains, 15 were selected as the optimised panel, while only 11% of serum samples showed positivity. A geographical variation in predominantly reactive serovars was observed, whereas reactivity was low with the saprophytic strain Patoc. Testing with paired sera yielded a higher sensitivity but provided only a retrospective diagnosis. Serological tests based on ELISA with complementary molecular diagnosis using PCR are a feasible and robust alternative approach to diagnose leptospirosis in countries having a higher burden of the disease.

## Author summary

Microscopic agglutination test is the most commonly used serological test in the diagnosis of leptospirosis. The test uses a live panel of *Leptospira* representing main serogroups and for proper performance of the test, an optimised panel which react well with patient samples need to be selected. This paper describes the procedure of selecting such an optimised panel for Sri Lanka which is a country hyper-endemic for leptospirosis. The test was done

**Data Availability Statement:** All relevant data are within the manuscript.

**Funding:** JV, SA and MM received a grant from The National Institute of Allergy and Infectious Diseases of the National Institutes of Health, Award Number U19AI115658. URL of funder: https://www.niaid.nih.gov. The content is solely the responsibility of the authors and does not necessarily represent the official views of the National Institutes of Health. The funders had no role in study design, data collection and analysis, decision to publish, or preparation of the manuscript.

**Competing interests:** The authors have declared that no competing interests exist.

on serum samples collected from patients presented with acute undifferentiated fever with 24 panel of serogroups and 15 were selected as the optimised panel. The test was found to have a low sensitivity in the acute stage and ELISA based serological tests with molecular diagnosis using PCR assays would be a better way for diagnosis of leptospirosis.

## Introduction

Leptospirosis, caused by a group of spirochetes in the family *Leptospiraceae*, is considered to be the most common zoonotic disease worldwide; with many mammalian species, mainly rodents, acting as reservoir hosts[1–3].Genomic classification of *Leptospira* has identified 64 species to date, with 17 in the type 1 pathogenic group which are responsible for the majority of disease cases[4]. *Leptospira interrogans*, *Leptospira borgpetersenii*, and *Leptospira kirschneri* are the predominant circulating species worldwide[5–7]. The serologic classification divides the strains into serovars according to the lipopolysaccharide structure of the outer cell wall; with more than 300 serovars grouped into 25 serogroups[3,8,9]. The importance of serologic classification is appreciated in part due to the established links with particular serovars and reservoir animals; for example, the serovar Icterohaemorrhagiae and Ballum in rodents and serovar Hardjo and Pomona in cattle[10,11].

The lack of a rapid and accurate diagnostic test in the acute stage of the disease is a major challenge for the management of leptospirosis. A further complication is the similar clinical manifestations of dengue fever, rickettsial infections, and malaria; which are common in leptospirosis endemic regions[12,13]. However, to diagnose leptospirosis clinicians mainly depend on clinical features combined with a history suggestive of exposure to a susceptible event. Unlike many other bacterial infections, culture isolation is not an option in the acute setting due to the fastidious nature of the organism, which needs special culture media and might take a minimum of 3 weeks to give a positive result[7,14].

The microscopic agglutination test (MAT) has long been in use as the serological reference field test of leptospirosis[15–19]. MAT depends on the immune response of the patient with the production of antibodies which are specific to the infecting serogroup. According to the criteria made by the "Leptospirosis Reference Epidemiology Group" (LERG), a confirmed case of leptospirosis should have an acute MAT titre of ≥1 in 400 or a four-fold rise of titre between acute and convalescent samples. A probable case is defined when the MAT titre is ≥1 in 100 in a non-endemic area [20]. Although MAT results mainly provide a retrospective diagnosis, the test serves to diagnose cases mainly for epidemiological purposes. This is especially important in estimating the true disease burden, and some studies have used MAT data to predict circulating serogroups within possible reservoir hosts[21–24].

The World Health Organisation (WHO) recommends to use a locally optimised MAT panel that represents the currently circulating strains in a particular region; or to use a broad panel of serogroups in the absence of such knowledge[11]. The basis for this is to improve the sensitivity of the test, as patient sera are likely to react well with local strains. However, knowledge on currently circulating strains is scarce in many high endemic settings. This is particularly a challenge given the laborious, low sensitive and resource intensive procedure involved in culture isolation of locally prevalent strains. Even the use of strains representative of a broad panel of serogroups for MAT is not feasible given the resource intensity and expense of the procedure.

The lack of a simple diagnostic test with higher sensitivity and specificity during the acute phase of the illness is a major challenge in managing patients with leptospirosis; and this could

have an impact on optimal patient care. Currently, molecular based diagnostic tests are rapidly replacing culture and serology based assays in diagnosing leptospirosis. Several qPCR assays utilizing pathogen specific primers have been assessed and found to have a higher sensitivity during the acute phase; a time point at which the serological response is lagging [25,26]. These molecular tests have the added advantage of being able to detect positive samples with non-viable organisms, such as when culture techniques fail following antibiotic therapy. In addition, there are published reports of several molecular based platforms which have been developed to determine the infecting species directly from the blood sample, without the need of culture [27,28].

ELISA and lateral flow based serological tests have largely replaced the conventional MAT test due to their ease of performance with comparable sensitivity and specificity, particularly during the acute stage of the disease[15,18,29–31]. Some of these serological tests have gained the status of point of care rapid screening tests[19,32–35].

Sri Lanka has a high incidence of leptospirosis with several recent outbreaks [1,36–38], and still faces challenges related to diagnostics to be used in the acute stage as well as selecting a locally optimised MAT panel. The knowledge on currently circulating strains is low, with a single recent report on two isolates[39]. However, MAT is used as the serological standard test in Sri Lanka and optimisation of the panel is an essential prerequisite to improve the sensitivity of the test. The purpose of the present study was to describe the procedure for selecting a sensitive as well as cost effective MAT panel in Sri Lanka using a cohort of patients with acute undifferentiated fever. This study might serve as an example for settings where there are knowledge gaps on circulating *Leptospira* serogroups.

## Methodology

### Ethics statement

Written informed consent was obtained from all patients prior to sample collection. For minors, written informed consent was obtained from the parents/ guardians. This study is approved by the Ethics Review Committee of the Faculty of Medicine and Allied Sciences, Rajarata University of Sri Lanka. Protocol No.ERC/2015/18.

### Study design

This study is part of a large multi-centre study to characterise the clinical, epidemiological, and aetiological aspects of leptospirosis in Sri Lanka. The study protocol is published elsewhere [40].

### Study setting

The study sites include a wide range of geographic areas differing in temperature, altitude, rainfall, ecology, human behaviour, and leptospirosis endemicity. Data were collected from June 2016 to January 2019.

The main data collection sites for the serological study were the Teaching Hospital Anuradhapura (THA) and Teaching Hospital Peradeniya (THP). A short-term data collection was done at the Base Hospital Awissawella (BHA) and Provincial General Hospital Rathnapura (PGHR), following flooding in 2017. These study sites represent four districts within four provinces of the country. In addition, the geographical distribution represents dry and wet zones, and low and high altitudes.

## Sample collection

Blood was collected from two types of patients. Firstly, acute undifferentiated febrile (AUF) patients (temperature >38˚C, fever <15 days) from outpatient departments and hospital wards. Patients from paediatric wards (age less than 12 years) were not included in this set of patients. Also excluded were physician-diagnosed cases of fever due to other causes, such as probable or definite acute bacterial meningitis, lower respiratory tract infections (e.g., consolidated lobar pneumonia), traumatic, post-operative, or fever due to nosocomial infections, and any patient confirmed with other diagnosis as a cause for the fever. The second type of patients were probable cases of clinical leptospirosis from any ward, without a restriction of age or fever duration. Most of the probable cases were actually detected as acute undifferentiated febrile patients for the first group. However, there were patient who were missed during the initial recruitment and or developed those symptoms later and the physicians wanted to exclude leptospirosis as an alternative diagnosis.

## Patient recruitment

The first category of patients was recruited on admission, by visiting all selected hospitals on a daily basis. A clinical data collector (graduate nurse or physician) screened the patients using the eligibility criteria and patients were recruited. Sample and data collection was done on the first day of admission. The second group of patients were recruited as reported.

## Sample processing and MAT procedure

From the eligible patients, 4 ml of blood was collected into plain tubes, allowed to coagulate for 30 minutes, and then centrifuged at 1300 rpm for 10 minutes. The serum was separated and 500 µl aliquots were prepared and stored at -20˚C and -80˚C for short term and long-term storage, respectively. The procedure of serum separation and storage was carried out within 2 hours of sample collection. A follow up serum sample was collected three weeks from the onset of fever and similarly prepared and stored.

The MAT panel selected for the study consists of WHO recommended serovars and is used by the Centre for Disease Control (CDC)[11]. Five additional CDC strains were included, which were isolated from the wet zone of Sri Lanka during the period 1950–1970 [36,41]. All strains were maintained in EMJH liquid media and sub-cultured weekly to maintain the live antigen panel. Cultures which were 4–5 days old and showed a growth equivalent to a 0.5 McFarland solution were selected for the study. For the MAT testing, two strains from the original panel were not included due to inadequate growth in the EMJH liquid media. These were *L. borgpetersenii* serovar Javanica str. Veldrat Bataviae and *L. interrogans* serovar Cynopteri str. 3522C. The final MAT panel consisted of 24 strains representing 17 serogroups which included a saprophytic species *Leptospira biflexa* serogroup Semaranga serovar Patoc.

MAT was run in two steps; screening and run out tests. In the first step serum samples were screened in 1 in 50 dilution by diluting with phosphate buffered saline (PBS) and screened in 96 well flat bottom microtitre plates with the panel of 24 live antigens. The first row of each plate contained antigen controls which were loaded in 50 µl of PBS and 50 µl of a live antigen strain making a final volume of 100 µl. The remaining rows were dedicated for a single serum sample and each column for a single strain. These wells were loaded with 50 µl of diluted serum (dilution of 1 in 25) and 50 µl of a specific live antigen making a final dilution of 1 in 50. An internal quality control step was carried out daily with each test run. Ideally, standard antisera should be available for each test strain for this purpose. However, it was available only for the *L. interrogans* serovar Weerasinghe str. Weerasinghe (purchased from the reference laboratory Amsterdam, Netherland). The standard antisera was mixed with the corresponding

serovar plus a distinct serovar in the panel to make sure that only specific agglutination reactions are occurring with appropriate test conditions.

After loading of samples and test strains, the plates were put into an orbital plate shaker for 5 minutes allowing optimal mixing of sera and live antigens. This was followed by incubation for two hours in a 30˚C incubator.

Agglutination reactions were read by adding 5 μl drops onto a clean glass slide from each well under dark field microscopy with 200x magnification. Positive samples were selected as those which gave 50% reduction of free, motile *Leptospira* compared to the antigen control. Reading was done column-wise where reaction to a single serovar was tested at a time. When paired sera were available, both samples were tested in the same plate.

In the second run out test, reactive sera via screening were tested against respective serovars with a dilution series from 1 in 5o to 1 in 3200 to detect the maximum dilution which yielded a 50% reduction of free, motile *Leptospira*. A positive MAT test was defined as seroconversion, a four-fold increase in titre between acute and convalescent samples or an acute titre of >1 in 400.

## Results

### Patient characteristics

From 2016–2018, we received 1132 samples from 982 patients for MAT. These samples were from four hospitals representing four districts (Table 1) and the majority were from inward patients (n = 904, 92%). Of these, 832 were received as single samples mostly during the acute illness and 150 as paired samples in the acute and convalescent periods. Patient ages ranged from 12 to 87 years. The median duration of fever was 4 days (range 2–5 days).

Summary of the MAT results is given in Table 2. Of the 150 paired samples, 19 (13%) had titres of ≥1 in 400 during the acute stage and another 11 (7%) samples had titres ranging from 1 in 50 to 1 in 200. Paired sample analysis showed 49 (33%) confirmed cases; with 22 (15%) sero-conversions(from non reactive to 1/50–2, to 1/200-2 and to ≥1/400-18), 12 (8%) with four-fold rise of titres, and another 15 (10%) with titres ≥1 in 400 in the acute stage but didn't show a four fold rise in the convalescent sample. Of the 832 single samples, 58 (7%) had titres ≥1 in 400.

To understand the MAT reactivity with all *Leptospira* strains used in the study, patient sample reactivity for individual antigens/strains in the MAT panel (irrespective of a single sample or paired samples) were screened. As shown in Table 3, all 24 *Leptospira* strains reacted with the MAT panel.

This strain level analysis shows a large number of reactive antigens which could be used for the diagnosis of leptospirosis in Sri Lanka. However, the cross reactions were high, especially in the samples with high titres (Fig 1), and some antigens seemed to yield consistently low titres. In samples with a titre of >1:3200, around 50% of the samples had cross reactivity across more than 10 antigens.

We characterized the antigens providing the highest titres for particular samples and titres ≥ 1:400 (Table 4) to select the best panel of *Leptospira* for the Sri Lankan MAT panel.

This analysis shows that, although the reactivity is high, for diagnostic purposes more than 95% sensitivity could be achieved using 11 antigens and 100% with 15 antigens, compared with the full 24 antigen panel. The selected panel included three Sri Lankan isolates plus the genus specific saprophyte Patoc strain.

### Varying highest titre based on day of sampling

Of the 19 paired samples with positive results, 15 had a titre of ≥1 in 400 in the acute samples with a higher titre in the convalescent period. We examined these samples to determine the serovar/serogroup prediction using the highest titre. Of the 15 samples, 12 (80%) had highest

**Table 1. Characteristics of the patient samples.**

|  | n | % |
|---|---|---|
| *Age* | | |
| <20 | 65 | 7 |
| 20–29 | 139 | 15 |
| 30–39 | 179 | 20 |
| 40–49 | 202 | 23 |
| 50–59 | 175 | 20 |
| 60–69 | 90 | 10 |
| ≥70 | 27 | 3 |
| *Sex* | | |
| Male | 722 | 80 |
| Female | 181 | 20 |
| *Ethnicity* | | |
| Sinhala | 842 | 93 |
| Sri Lankan Moor/ Malay | 27 | 3 |
| Sri Lankan Tamil | 32 | 3 |
| Other | 3 | 0.3 |
| *Hospital/District* | | |
| ABH | 149 | 15 |
| RGH | 72 | 7 |
| THA | 513 | 52 |
| THP | 248 | 25 |
| *Patient presentation* | | |
| Outpatient | 71 | 7 |
| Hospitalized | 904 | 92 |
| *Clinical presentation* | | |
| Three classical* only | 382 | 50 |
| Three classical and jaundice or conjunctival suffusion | 156 | 20 |
| Three classical and jaundice and conjunctival suffusion | 28 | 3 |
| Any two classical features out of three and jaundice/conjunctival suffusion | 32 | 4 |
| Other | 62 | 8 |
| No clinical details | 95 | 12 |

*Three classical features: headache, myalgia, and fever

titres for different antigens in the convalescent samples, showing that the routine diagnostic MAT based serovar or serogroup prediction is unreliable and may depend heavily on the day of sampling. Fig 2 and Table 5 show reactivity of patient sera in different geographical settings.

**Table 2. Final diagnosis for 982 patients based on the MAT results.**

|  |  | n | % |
|---|---|---|---|
| Confirmed 107 (11%) | Seroconversion | 22 | 2 |
|  | Fourfold rise | 12 | 1 |
|  | Single titre ≥1 in 400 | 73 | 8 |
| Reactive/Probable | Reactive (≥50 titre <1 in 400) | 50 | 5 |
| Negative 825 (84%) | Non-reactive paired sample | 98 | 10 |
|  | Non-reactive single sample | 727 | 74 |
| Total |  | 982 | 100 |

**Table 3. Highest titres of 1132 serum samples to individual antigens used in the MAT panel.**

| Strain | 1 in 50 | 1 in 100 | 1 in 200 | 1 in 400 | 1 in 800 | 1 in 1600 | 1 in 3200 | >1 in 3200 | Total reactive samples |
|---|---|---|---|---|---|---|---|---|---|
| *L. interrogans* serogroup Australis serovar Bratislava str. Jez-Bratislava | 13 | 8 | 12 | 17 | 8 | 8 | 15 | 3 | 84 |
| *L. interrogans* serogroup Canicola serovar Canicola str. Ruebush | 10 | 6 | 13 | 14 | 10 | 3 | 8 | 3 | 67 |
| *L. interrogans* serogroup Autumnalis serovar Weerasinghe str. Weerasinghe | 9 | 14 | 12 | 13 | 4 | 1 | 0 | 5 | 58 |
| *L. interrogans* serogroup Icterohaemorrhagiae serovar Icterohaemorrhagiae str. RGA | 2 | 4 | 11 | 8 | 9 | 7 | 6 | 3 | 50 |
| *L. santarosai* serogroup Mini serovar Georgia str. LT 117 | 6 | 13 | 13 | 9 | 2 | 0 | 0 | 0 | 43 |
| *L. interrogans* serogroup Bataviae serovar Bataviae str. Van Tienen | 0 | 3 | 5 | 8 | 10 | 5 | 5 | 0 | 36 |
| *L. interrogans* serogroup Icterohaemorrhagiae serovar Mankarso str. Mankarso | 2 | 5 | 5 | 4 | 4 | 5 | 1 | 2 | 28 |
| *L. biflexa* serogroup Semaranga serovar Patoc strain Patoc1 | 6 | 2 | 4 | 7 | 4 | 1 | 1 | 0 | 25 |
| *L. interrogans* serogroup Pyrogenes serovar Alexi str. 616 | 6 | 7 | 8 | 1 | 0 | 0 | 1 | 0 | 23 |
| *L. santarosai* serogroup Pyrogenes serovar Pyrogenes str. Salinem | 0 | 4 | 2 | 4 | 7 | 3 | 0 | 1 | 21 |
| *L. borgpetersenii* serogroup Javanica serovar Ceylonica str. Piyasena | 4 | 1 | 4 | 6 | 2 | 2 | 0 | 0 | 19 |
| *L. interrogans* serogroup Australis serovar Australis str. Ballico | 7 | 2 | 5 | 1 | 2 | 1 | 1 | 0 | 19 |
| *L. weilii* serogroup Celledoni *serovar* Celledoni str. Celledoni | 7 | 2 | 0 | 3 | 2 | 1 | 0 | 0 | 15 |
| *L. interrogans* serogroup Sejroe *serovar* wolfii str. 3705 | 0 | 3 | 4 | 4 | 0 | 1 | 0 | 0 | 12 |
| *L. kirschneri* serogroup Grippotyphosa serovar Ratnapura str. Wumalasena | 4 | 3 | 4 | 0 | 1 | 0 | 0 | 0 | 12 |
| *L. interrogans* serogroup Autumnalis serovar Autumnalis str. Akiyami A | 2 | 0 | 3 | 3 | 1 | 1 | 0 | 0 | 10 |
| *L. interrogans* serogroup Djasiman serovar Djasiman str. Djasiman | 1 | 2 | 2 | 2 | 0 | 2 | 0 | 0 | 9 |
| *L. borgpetersenii* serogroup Ballum serovar Ballum str. Mus 127 | 1 | 4 | 3 | 0 | 0 | 0 | 0 | 0 | 8 |
| *L. borgpetersenii* serogroup Tarassovi serovar tarassovi str. Perepelitsyn | 3 | 4 | 1 | 0 | 0 | 0 | 0 | 0 | 8 |
| *L. santarosai* serogroup Autumnalis serovar Alice str. Alice | 1 | 1 | 1 | 0 | 2 | 0 | 0 | 0 | 5 |
| *L. interrogans* serogroup Grippotyphosa serovar Grippotyphosa | 0 | 1 | 0 | 2 | 1 | 0 | 0 | 0 | 4 |
| *L. interrogans* serogroup Sejroe serovar Geyaweera str. Geyaweera | 0 | 0 | 0 | 0 | 1 | 0 | 0 | 1 | 2 |
| *L. santarosai* serogroup Hebdomadis serovar Borincana str. HS 622 | 2 | 0 | 0 | 0 | 0 | 0 | 0 | 0 | 2 |
| *L. interrogans* serogroup Pomona serovar Pomona str. Pomona | 0 | 0 | 0 | 0 | 0 | 1 | 0 | 0 | 1 |

## Discussion

Although MAT is considered to be the serological reference test for the diagnosis of leptospirosis, it has many inherent issues in performance. The inability to standardize the test conditions may lead to interpersonal variations in the interpretation of results. The necessity to maintain live cultures for the MAT panel is a laborious procedure due to the fastidious nature of the species, and imposes a risk of laboratory acquired infections. Apart from these technical problems, the sensitivity of MAT in the acute stage has been questioned. Patients might not elicit detectable immune responses in the acute stage and sensitivity can be significantly low in such cases [32,42–44]. Therefore, for a proper interpretation, paired sera should be used[11][32]. However, relatively higher specificity of MAT has been reported in both acute and paired sera for the diagnosis [32,33,42].

The findings of this study have several important implications. This study used a broad panel of 24 strains representing 17 serogroups per WHO recommendation, in the absence of recent published data on widely circulating local strains. This is the recommended practice for settings where knowledge on current circulating serovars is missing. Our results show that in

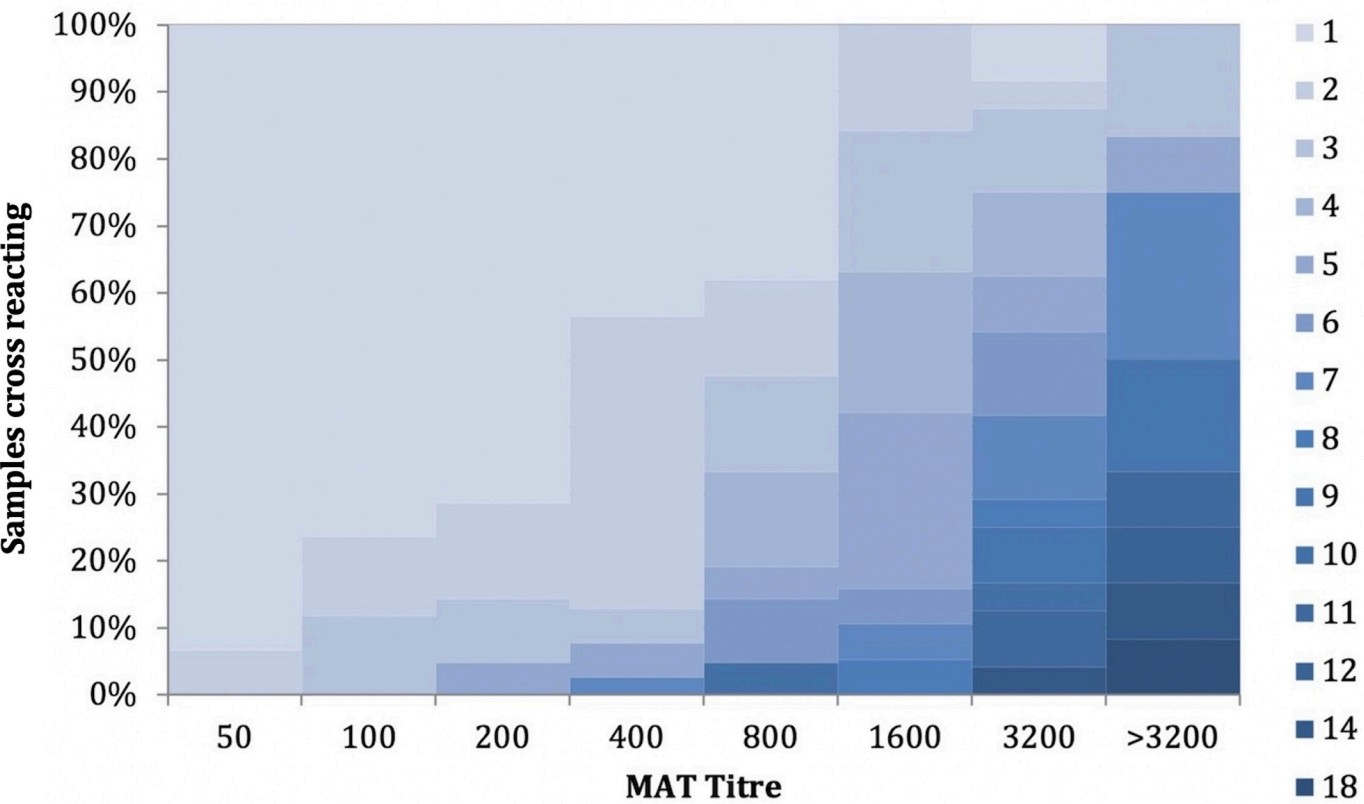

**Fig 1. Percentage distribution of cross reactivity of patient samples by the highest MAT titre.** The colour legends 1–18 shows the number of strains reacting with a single serum sample.

Sri Lankan settings, 100% sensitivity of MAT could be achieved using 15 rather than the recommended 24 strains. These 15 strains compose a locally optimised panel since the reactive sera were collected from patients residing in different geographical regions, including dry and wet zones and high and low altitudes. Although the hospitals were from four provinces, the actual patient population was from eight of nine provinces. Furthermore, the samples were collected over a 2.5 year period where both endemic and epidemic cases were included in the study.

In this study, differential patterns of sera reactivity were observed in the diverse geographical settings. Predominant reactivity only with serogroup Australis serovar Bratislava in the dry zone contrasts with the shared reactivity with several serovars in the wet zone. This might reflect the diversity of reservoir hosts in different geographical contexts. Some studies have used MAT data to predict infecting serogroups[45]; however, this has been shown to be inaccurate due to cross reactions with other serovars in the acute stage and is not generally recommended[11,46,47]. This study similarly revealed such multiple cross reactions, particularly with sera at high titres.

In comparison, published MAT data using an almost similar representative serogroup panel in a 2008 leptospirosis outbreak in Sri Lanka revealed *L. interrogans* serogroup Pyrogenes serovar Pyrogenes to be the predominant reactive serovar[48]. The shift in the reactive serovar from Pyrogenes to Bratislava in this study may also be indirect evidence of changes in predominant reservoir hosts over time. It may reflect the predominant role of one serovar in an outbreak setting and geographical difference in circulating serovars. The 2008 study was

**Table 4. Number of samples with diagnostic (≥ 1 in 400) and highest tires with individual antigens for 1132 patient sera.**

| Strain | Titres ≥1 in 400 | Highest titre for the sample | Diagnostic and highest titre | Cumulative percentage of diagnosed samples |
|---|---|---|---|---|
| *L. interrogans* serovar Bratislava str. Jez-Bratislava | 51 | 70 | 46 | 34% |
| *L. interrogans* serovar Canicola str. Ruebush | 38 | 35 | 20 | 49% |
| *L. interrogans* serovar Icterohaemorrhagiae str. RGA | 33 | 17 | 15 | 60% |
| *L.interrogans* serovar Weerasinghe str. Weerasinghe | 23 | 21 | 15 | 71% |
| *L. interrogans* serovar Bataviae str. Van Tienen | 28 | 12 | 9 | 78% |
| *L. santarosai* serovar Pyrogenes str. Salinem | 15 | 5 | 5 | 82% |
| *L. interrogans* serovar wolfii str. 3705 | 5 | 6 | 5 | 85% |
| *L. interrogans* serovar Mankarso str. Mankarso | 16 | 4 | 4 | 88% |
| *L. borgpetersenii* serovar Ceylonica str. Piyasena | 10 | 8 | 4 | 91% |
| *L. santarosai* serovar Georgia str. LT 117 | 11 | 8 | 3 | 94% |
| *L. biflexa* serovar Patoc strain Patoc1 | 13 | 5 | 2 | 95% |
| *L. interrogans* serovar Australis str. Ballico | 5 | 2 | 2 | 97% |
| *L. interrogans* serovar Geyaweera str. Geyaweera | 2 | 2 | 2 | 98% |
| *L. interrogans* serovar Djasiman str. Djasiman | 4 | 3 | 1 | 99% |
| *L. interrogans* serovar Pomona str. Pomona | 1 | 1 | 1 | 100% |
| *L. weilii* serovar Celledoni str. Celledoni | 6 | 0 | 0 | |
| *L. interrogans* serovar Autumnalis str. Akiyami A | 5 | 1 | 0 | |
| *L.interrogans* serovar Grippotyphosa | 3 | 0 | 0 | |
| *L. interrogans* serovar Alexi str. 616 | 2 | 1 | 0 | |
| *L. santarosai* serovar Alice str. Alice | 2 | 0 | 0 | |
| *L. kirschneri* serovar Ratnapura str. Wumalasena | 1 | 0 | 0 | |
| *L. borgpetersenii* serovar Ballum str. Mus 127 | 0 | 1 | 0 | |
| *L. borgpetersenii* serovar tarassovi str. Perepelitsyn | 0 | 1 | 0 | |
| *L. santarosai* serovar Borincana str. HS 622 | 0 | 0 | 0 | |

focussed in a wet zone whereas this study included both dry and wet zones, with predominant reactivity in samples from the dry zone. However, support for this would be to simultaneously recover local isolates from humans and veterinary animals where serotyping can be used to establish important and potential reservoir hosts for human leptospirosis.

We did not observe an anticipated high reactivity with the local isolates in the MAT panel. Only three isolates were among the 15 serovars of the selected final panel: serovars Weerasinghe, Ceylonica, and Geyaweera. These three isolates were recovered from the wet zone of Sri Lanka during 1964–1965[41]. The analysis of agglutination reactions with local serovars shows that only serovar Weerasinghe was reactive with 37% of the total reactive samples whereas that of serovar Ceylonica and serovar Geyaweera showed only 12% and 1% reactivity, respectively. These observations follow a similar pattern to MAT data published in the 2008 Sri Lanka study[48]. Changes in antigenic structures or emergence of new serovars over time might have contributed to this reaction pattern. Similar results have been found in other international studies where local isolates have failed to give better agglutination reactions[49]. Inclusion of an adequate number of currently circulating *Leptospira* isolates in the MAT panel and a re-evaluation procedure is required before committing to the use of new isolates. The

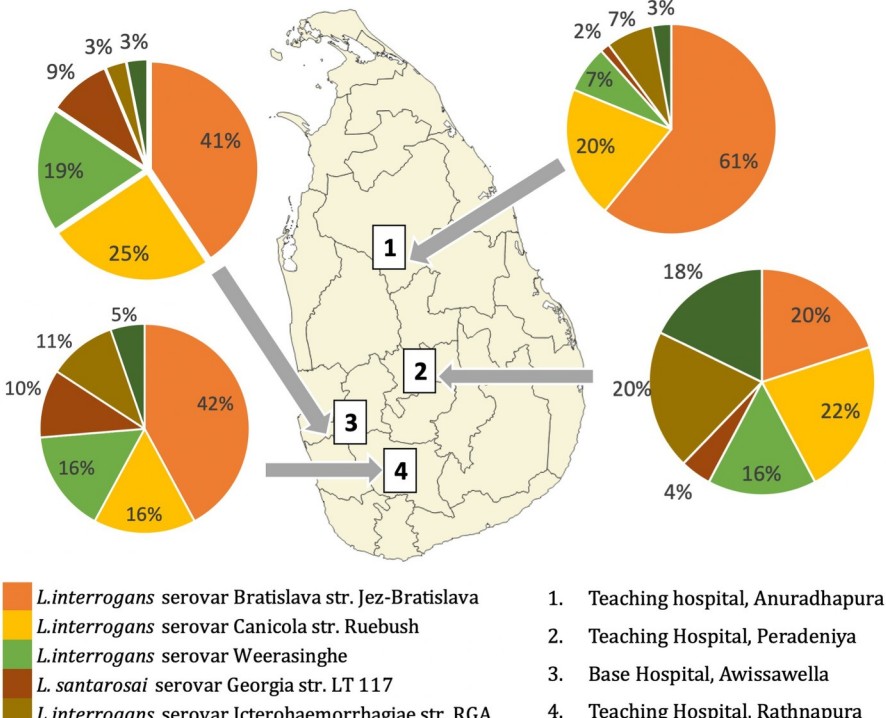

**Fig 2. Reactivity of patient sera from different geographical settings in Sri Lanka. (Maps used for the baselayer in Fig 2 are freely available from The United Nations Office for the Coordination of Humanitarian Affairs https:// data.humdata.org/dataset/sri-lanka-administrative-levels-0-4-boundaries**). Serological reactions differed across the four hospitals used in the study. We looked at only those strains with high titres (Fig 2). Peradeniya (wet zone high lands) showed high titres to at least five strains, while Anuradhapura patients were predominantly reacting against only one strain. The observed geographical differences of *L. interrogans* serovar Bratislava str. Jez-Bratislava, *L. interrogans* serovar Icterohaemorrhagiae str. RGA and *L. interrogans* serovar Bataviae str. Van Tienen in reactivity were statistically significant (chi-square 34.1, p = .04).

on-going studies in Sri Lanka with recently recovered isolates will provide adequate new strains for this purpose[50]

A relatively low reactivity was observed with the saprophytic non-pathogenic serovar Patoc (16%). Usually the saprophytic species *L. biflexa* is considered to have a broader reactivity and is therefore recommended to be included in MAT panels [11]. However, our results were consistent with an ELISA-based study in Sri Lanka which showed reduced sensitivity of saprophyte *L. biflexa* in serological assays compared to a pathogenic local isolate [51]. Similar results

**Table 5. Reactivity of patient sera from different geographical settings.**

|  | ABH | | RGH | | THA | | THP | |
|---|---|---|---|---|---|---|---|---|
|  | n | % | n | % | n | % | n | % |
| *L. interrogans* serovar Bratislava str. Jez-Bratislava | 13 | 38 | 8 | 38 | 42 | 51 | 9 | 19 |
| *L. interrogans* serovar Canicola str. Ruebush | 8 | 23 | 3 | 14 | 14 | 17 | 10 | 21 |
| *L. interrogans* serovar Weerasinghe | 6 | 17 | 3 | 14 | 5 | 6 | 7 | 15 |
| *L. santarosai* serovar Georgia str. LT 117 | 3 | 8 | 2 | 9 | 1 | 1 | 2 | 4 |
| *L. interrogans* serovar Icterohaemorrhagiae str. RGA | 1 | 2 | 2 | 9 | 5 | 6 | 9 | 19 |
| *L. interrogans* serovar Bataviae str. Van Tienen | 1 | 2 | 1 | 4 | 2 | 2 | 8 | 17 |

have been observed in studies done in other countries with the Patoc strain[52]. Based on these observations, MAT should be performed with a panel of serovars; and testing only with saprophytic *L. biflexa*, assuming a broader range of reactivity, could lead to gross underestimation of leptospirosis cases.

This study has few technical shortcomings. Ideally, for the internal quality control step, antisera should be available for each strain in the test panel and should be run with each test. In addition to ensuring quality control of the test, this step will also enable identifying the mislabelling of live antigens which might happen during long-term maintenance. However, due to financial issues the purchasing of antisera for the whole panel wasn't possible, and hence only one antiserum was used as a quality control.

We observed that a large number of "clinical leptospirosis" patients were negative. This low sensitivity could be due to several possibilities. Ideally, before concluding a negative result, paired sera should be available to look for seroconversion or a four-fold rise in titre. However, the lack of paired sera for a majority of samples (available only for 15%) in this study might contribute to a low level of detection. The findings of this study also highlight the poor utility of MAT as a diagnostic tool in the acute stage of the disease which is the critical period in patient management.

Another possibility for the low positivity could be due to other aetiologies in these patients such as simple viral fever, dengue fever or rickettsiosis as they are from a cohort of acute undifferentiated fever patients. Of the patients we tested, 51% had only fever, headache, and myalgia —a set of symptoms which may appear due to many tropical fevers.

On other other hand, some of the patients confirmed as having a single high titre may probably be affected by high background tires. There are no Sri Lankan studies estimating background titres and it may slightly overestimate the positivity. However, it will not affect the main focus of the study.

MAT is used as the serological reference test despite having low sensitivity, low value as a clinical diagnostic tool, and involving labour intense procedures. Hence determining the optimum number of strains to be included into the test panel is a crucial step in many settings where the resources are limited. For Sri Lanka, further improvement of the proposed panel is required with the addition of newly isolated strains.

Considering the laborious nature of the test procedure and low sensitivity in the acute stage, the utility of MAT as a reference test seems to be imprudent. Rapid bed side diagnostics are a reasonable alternative to overcome the inherent issues with MAT. These diagnostics include lateral flow immune assays as screening tests, supported with ELISA-based tests as confirmatory assays. However ELISA based serological studies also need to be validated as there can be many cross reactions with other infections. Furthermore the antigen should be regionally optimised to increase the sensitivity of the test as has been shown in studies done in the local setting[51]. A highly sensitive and specific serological tests like immunofluroscence assays haven't been extensively studied limiting its' use in the resource poor endemic regions [20] Although a PCR-based molecular diagnostic is expensive and technically demanding, it would be a good complimentary test to be used in the early stage of the disease before a serological response develops. Ideally the use of MAT should be preserved for epidemiological purposes, especially in endemic countries.

## Acknowledgments

We would like to thank Ms. Thilakanjali Gamage, Mr. K.M.R. Premathilaka, Mr. S.K. Senevirathna, and Mr. Milinda Perera for technical assistance, Mr. Shalka Srimantha and Ms. Chamila Kappagoda for their extensive support in culture maintenance and laboratory work. We also

thank all the physicians and healthcare staff in the various participating hospitals for the extended support given throughout this study.

## Author Contributions

**Conceptualization:** Dinesha Jayasundara, Joseph M. Vinetz, Suneth Agampodi.

**Data curation:** Dinesha Jayasundara, Janith Warnasekara, Suneth Agampodi.

**Formal analysis:** Dinesha Jayasundara, Indika Senavirathna, Janith Warnasekara.

**Funding acquisition:** Joseph M. Vinetz, Suneth Agampodi.

**Investigation:** Dinesha Jayasundara, Indika Senavirathna.

**Methodology:** Dinesha Jayasundara, Chandika Gamage, Indika Senavirathna, Joseph M. Vinetz, Suneth Agampodi.

**Project administration:** Suneth Agampodi.

**Supervision:** Chandika Gamage, Michael A. Matthias, Joseph M. Vinetz, Suneth Agampodi.

**Visualization:** Dinesha Jayasundara.

**Writing – original draft:** Dinesha Jayasundara.

**Writing – review & editing:** Chandika Gamage, Janith Warnasekara, Michael A. Matthias, Joseph M. Vinetz, Suneth Agampodi.

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
