## [Decision Letter · Decision Letter 0]

5 May 2021

Dear Dr. Suneth Agampodi,

Thank you very much for submitting your manuscript "Optimizing the microscopic agglutination test (MAT) panel for the diagnosis of Leptospirosis in a low resource, hyper-endemic setting with varied microgeographic variation in reactivity." for consideration at PLOS Neglected Tropical Diseases. As with all papers reviewed by the journal, your manuscript was reviewed by members of the editorial board and by several independent reviewers. The reviewers appreciated the attention to an important topic. Based on the reviews, we are likely to accept this manuscript for publication, providing that you modify the manuscript according to the review recommendations. 

Sincerely,

Vasantha kumari Neela

Associate Editor

Amanda Bastos

Deputy Editor

Reviewer's Responses to Questions

**Key Review Criteria Required for Acceptance?**

**Methods**

-Are the objectives of the study clearly articulated with a clear testable hypothesis stated?

-Is the study design appropriate to address the stated objectives?

-Is the population clearly described and appropriate for the hypothesis being tested?

-Is the sample size sufficient to ensure adequate power to address the hypothesis being tested?

-Were correct statistical analysis used to support conclusions?

-Are there concerns about ethical or regulatory requirements being met?

Reviewer #1: (No Response)

Reviewer #2: General Observations:

The microscopic agglutination test (MAT) is the basis or cornerstone test for serological classification and diagnosis. However, in the recent past MAT is used for the sero- epidemiological studies than diagnosis. MAT antibodies appear at the end of the first week of the onset of the disease, reaches to peak during the second week or early third week. Hence detecting MAT antibodies using MAT may not much useful for the clinical management of patients . More so single MAT may not provide conclusive information on current clinical infection or confirmatory diagnosis and technically cumbersome. Molecular genetic-based techniques with a conjunction of IgM ELISA (for early and late reporting cases respectively ) are in use to provide the highest patient care. Authors themselves expressed the similar opinion. Nevertheless, MAT is the reference test and the choice for serological classification and tracking of animal vectors. Optimizing the panel of serovars / strains for the microscopic agglutination test (MAT) for the diagnosis of Leptospirosis (use of representative strains/ serovars reported (one or two strains from each serogroup ) and inclusion local isolates is well known and not a new . More so geographic genomics - gene acquisition and gene loss on evolutionary time scale is a continues process in leptospires and it is evidence from classification – about 300 serovars and several species (sensu stricto ). In view of the above phenomenon , optimizing MAT panel at particular point of time may not be ideal or optimized panel at another point of time.

Specific points:

Introduction- Line 92 – 94: No single MAT titre can be regarded as diagnostic of acute or current clinical infection , in a proportion of confirmed patients may have low MAT titres or past infection may have a high titre ( it is well known that microscopic agglutinating antibodies are long-lasof ting and persist several months to several years ). More so , prevalence of microscopic agglutinating antibodies are commonly observed among the community , use of single MAT may increase the false positive results . Since the central dogma of serology is defined as four-fold rise in titre or greater rise in titres, demonstration of rising titres by using paired MAT only confirm the current infection . More so it is observed that the dilutions are mentioned as 1: 400 or 1: 800 or so on. Please be noted here that In the diagnostic serology, the dilutions should be expressed as 1 in 400 or 1in 800 or so on as 1 in 400 is different from 1: 400 . (1 part of serum and 399 parts diluent and 1 part of serum and 400 parts diluent respectively) . Can be please rewritten for better understanding.

Methodology : Line 183 -186 : Please see general observations - MAT panel for the diagnosis of leptospirosis and please clarify .

Reviewer #3: The theme was well contextualized in the introduction subsidizing the development of the manuscript. The importance this work has it’s from the optimization of MAT with the proposal to change the panel of serovars used mainly in the countries with limited resources, reduction of the number of strains tested without loss of reactivity and epidemiological assessment of the disease. The etiological agent, phenotypic and genotypic classification, diversity, occurrence and it interaction with different hosts were referenced; difficulty in early diagnosis (acute phase) of leptospirosis. The definition of cases for non-endemic areas was considered as proposed by “Leptospirosis Reference Epidemiology Group”. Also, complexity of its implementation, limitations and importance of MAT, as well as the advantages and application of the techniques available for the diagnosis of leptospirosis. 

The purpose of the study is clear. The study design was appropriate to achieve the objectives of the study. The authors will use samples from a cohort of patients with undifferentiated fever. These were divided into two types/groups: the first group of patients was with undifferentiated acute fever treated in outpatient department, hospital wards and paediatrics wards with ≥ 12 years old; the second group were patients with suspect symptoms of leptospirosis from any ward, without age restriction. The cohort study was from June 2016 to January 2019 in which the samples came from geographic areas with different characteristics (temperature, altitude, ecology, rainfall, human behavior and endemicity). The study observed the ethical requirements. 

Considering that Sri Lanka is a hyperendemic country for leptospirosis, is the cut-off point 1:100 or 1:50? Why was the cut-off point 1:100 used in the screening test while in titration it was used 1:50?

**Results**

-Does the analysis presented match the analysis plan?

-Are the results clearly and completely presented?

-Are the figures (Tables, Images) of sufficient quality for clarity?

Reviewer #1: (No Response)

Reviewer #2: Results : 

Lines 235-240: It is stated that 19 of the 150 patients with paired samples had a titre of 1 in 400 or more in the acute sample. Subsequently it is stated that 15 patients had a titre of 1 in 400 or more, but did not meet the other criteria for a positive diagnosis. I assume that 4 patients had an initial titre of 1 in 400 or more, but showed 4 fold rise in titre in the convalescent sample. If this is so, it may not be clear to the reader. Please state this clearly.

Lines 247 – 251: ‘We predicted the number of positives, if paired samples were available for all patients…’. Does this mean that the authors applied the proportion of positive cases among the 150 patients (33%) to the entire cohort of 982 patients to estimate the number of cases among them? If so, please clarify this and state the estimated number. 

Lines 249-251: Did the authors calculate the 95% confidence interval of the estimated number? What is 29%? Is it the width of the confidence interval of the estimated prevalence of leptospirosis among the 150 patients whose paired sera was available? Please state the confidence limits as lower and upper bounds.

Line 250-251: It is not clear what ‘difference’ was statistically significant. How McNemar test, which is for paired data, was performed? . Please clarify .

Line 253 , table 2 : A total of 107 patients confirmed as positives and out which 73 based on single MAT . Since the study was conducted in hyperendemic area out 73 confirmed patients , there is a probability that the proportion these 73 cases may be false positives (please see Specific points mentioned – Introduction -line 92 – 94 ) . The base line MAT titres in the community may vary from one geographical area to other . Is there any community based study conducted at the study area to estimate the base line titre and to draw cut -off titre for single MAT . Please clarify .

Reviewer #3: The characterization of patients does not refer to the two types (groups) of patients initially presented (lines 157 to 166). The clinical characteristics and the disease course are extremely important for results interpretation of MAT as well as other methodologies.

I recommend adding stratify the samples evaluated by MAT according to the disease course (days or weeks after the onset of symptoms).

 The authors used samples with less then 15 days. MAT shows reactivity from the 7th or 8th day of onset symptom and there is a considerable difference in sensitivity before and after the first week of illness. The analysis of the results considering the first and second week of illness could provide different information. This method is the gold standard recommended by the WHO as well as by other institutions and entities such as CDC and ILS. Was other methodology performed (PCR, ELISA) in parallel to MAT of the two types/groups of patients studied? Table 1 shows data from 982 samples from 1132 patients. Of the total samples, 105 patients did not report age, 79 not report sex, 78 ethnicity, 7 patients were lost without information about they were hospitalized or were on an outpatient departments and about the clinical presentation, 227 patients not even were mentioned if they presented other symptoms.

I recommend completing the information even when it has not been properly reported.

Under the table 1, line 237, the three classic characteristics (symptoms) are referenced: headache, myalgia and fever. I recommend that, in the same way, the two classic characteristics (symptoms) be referenced (myalgia and fever?). 

 The line 239 mention of the 150 paired samples, 19 (13%) had titres of ≥1:400 during the acute stage and lines 240-241 mention another 11 (7%) samples had titres ranging from 1:50 to 1:200. Paired sample analysis showed 49 (33%) confirmed cases. I can conclude that, of the 49 paired samples 19 it were from the convalescent phase?

I recommend conceptualize the acute phase by defining the period in days and rewriting the text from lines 239 to 244 with the exception of the last sentence to improve understanding.

 The writing of the line 243 is not clear “and another 15 (10%) with titres ≥1: 400 but not falling into the above two categories”.

 In Table 2, which type of patients according to the group mentioned in lines157 to 166 does the 107 leptospirosis confirmed patients belong to?

 Table 3 show that were used serovars of 17 serogroups recommended by the WHO, but there are different serovars and strains from the recommended panel. If were used the exactly recommended serovars and strains, could it generate a different reactivity profile? 

 Please review the wording for lines 239 and 290. The line 239 mention that is 19 positive paired samples while the line 290 mention 15 positive paired samples in the acute phase? Which is the correct number of sample?

**Conclusions**

-Are the conclusions supported by the data presented?

-Are the limitations of analysis clearly described?

-Do the authors discuss how these data can be helpful to advance our understanding of the topic under study?

-Is public health relevance addressed?

Reviewer #1: Despite the shortcomings of the MAT, well described here, also ELISA tests do have shortcomings (ie cross reactions). This also should be adressed in the discussion, to avoid that readers use any ELISA which tests for antileptospiral antibodies.

Reviewer #2: A total of 107 patients confirmed as positives and out which 73 based on single MAT . Since the study was conducted in hyperendemic area out 73 confirmed patients , there is a probability that the proportion these 73 cases may be false positives (please see Specific points mentioned – Introduction -line 92 – 94 ) . The base line MAT titres in the community may vary from one geographical area to other . Is there any community based study conducted at the study area to estimate the base line titre and to draw cut -off titre for single MAT . Please clarify .

Reviewer #3: Manuscript discussion emphasizes the difficulties in performing the MAT technique. Considering that, the objective of the article was “The purpose of the present study was to describe the procedure for selecting a sensitive as well as cost effective MAT panel in Sri Lanka using a cohort of patients with acute undifferentiated fever”.

I recommend including in the discussion the limitations of the other diagnostic methods, mainly the serological tests.

In this study, were the samples submitted to analysis by another methodology? If so, it would be important to present these data.

**Editorial and Data Presentation Modifications?**

Reviewer #1: 1. When reading the procedure of the MAT, it seems that the authors did not include the volume of the leptospiral antigen in the titers. They describe the serum sample is diluted 1:50. They also describe their dilution series ranging from 1:50 to 1:3200. In the results titres of 1:50 are described. To my knowledge most of the laboratories performing MAT report titers including the added antigen. I would suggest the authors to state cleary that they report the serum dilutions, without the antigen, so that this is clear to all readers. 

2. The sentence from line 257 to 258 needs to be reformulated.It seems the authors mean that from the panel of leptospiral strains, there was with each serovar a MAT titer with 1 or more of the 982 serum samples. Or they should say that by MAT panel they mean the sera tested by MAT. But earlier in the methods (line 183) they state that the MAT panel consists of leptospiral serovars.

3. strain Van Tienan should be spelled as Van Tienen.

Reviewer #2: As Suggested under methodology and results

Reviewer #3: (No Response)

**Summary and General Comments**

Reviewer #1: This is a clear study, which gives a good example about how to set up a MAT panel for a certain geographical area. Most important is to realize that a situation is not static and new serovar might appear in a region.

Reviewer #2: The microscopic agglutination test (MAT) is the basis or cornerstone test for serological classification and diagnosis. However, in the recent past MAT is used for the sero- epidemiological studies than diagnosis. MAT antibodies appear at the end of the first week of the onset of the disease, reaches to peak during the second week or early third week. Hence detecting MAT antibodies using MAT may not much useful for the clinical management of patients . More so single MAT may not provide conclusive information on current clinical infection or confirmatory diagnosis and technically cumbersome. Molecular genetic-based techniques with a conjunction of IgM ELISA (for early and late reporting cases respectively ) are in use to provide the highest patient care. Authors themselves expressed the similar opinion. Nevertheless, MAT is the reference test and the choice for serological classification and tracking of animal vectors. Optimizing the panel of serovars / strains for the microscopic agglutination test (MAT) for the diagnosis of Leptospirosis (use of representative strains/ serovars reported (one or two strains from each serogroup ) and inclusion local isolates is well known and not a new . More so geographic genomics - gene acquisition and gene loss on evolutionary time scale is a continues process in leptospires and it is evidence from classification – about 300 serovars and several species (sensu stricto ). In view of the above phenomenon , optimizing MAT panel at particular point of time may not be ideal or optimized panel at another point of time.

Reviewer #3: I congratulate the authors for the research developed. The article is relevant for the improvement of leptospirosis diagnosis. The leptospirosis is neglected in many countries because they are unable to diagnose it in a timely manner.

The MAT method is extremely labor and there is a need for the development of new methodologies that allow early diagnosis as well as the identification of the different serovars circulating in different countries.

Currently, MAT is important both for the diagnosis of leptospirosis and to verify the endemicity of the disease.

PLOS authors have the option to publish the peer review history of their article (what does this mean?). If published, this will include your full peer review and any attached files.

Reviewer #1: No

Reviewer #2: No

Reviewer #3: No

Figure Files:

Data Requirements:

Reproducibility:

References

---

## [Decision Letter · Decision Letter 1]

15 Jun 2021

Dear Dr. Suneth Agampodi,

We are pleased to inform you that your manuscript 'Optimizing the microscopic agglutination test (MAT) panel for the diagnosis of Leptospirosis in a low resource, hyper-endemic setting with varied microgeographic variation in reactivity.' has been provisionally accepted for publication in PLOS Neglected Tropical Diseases.

Best regards,

Vasantha kumari Neela

Associate Editor

Amanda Bastos

Deputy Editor

Reviewer's Responses to Questions

**Key Review Criteria Required for Acceptance?**

**Methods**

-Are the objectives of the study clearly articulated with a clear testable hypothesis stated?

-Is the study design appropriate to address the stated objectives?

-Is the population clearly described and appropriate for the hypothesis being tested?

-Is the sample size sufficient to ensure adequate power to address the hypothesis being tested?

-Were correct statistical analysis used to support conclusions?

-Are there concerns about ethical or regulatory requirements being met?

Reviewer #1: (No Response)

Reviewer #2: (No Response)

**Results**

-Does the analysis presented match the analysis plan?

-Are the results clearly and completely presented?

-Are the figures (Tables, Images) of sufficient quality for clarity?

Reviewer #1: (No Response)

Reviewer #2: (No Response)

**Conclusions**

-Are the conclusions supported by the data presented?

-Are the limitations of analysis clearly described?

-Do the authors discuss how these data can be helpful to advance our understanding of the topic under study?

-Is public health relevance addressed?

Reviewer #1: (No Response)

Reviewer #2: (No Response)

**Editorial and Data Presentation Modifications?**

Reviewer #1: I just noticed a small typo: In line 387 "On other rother hand"

Reviewer #2: (No Response)

**Summary and General Comments**

Reviewer #1: (No Response)

Reviewer #2: (No Response)

PLOS authors have the option to publish the peer review history of their article (what does this mean?). If published, this will include your full peer review and any attached files.

Reviewer #1: No

Reviewer #2: No

---

## [Editor Report · Acceptance letter]

28 Jun 2021

Dear Dr Agampodi,

We are delighted to inform you that your manuscript, "Optimizing the microscopic agglutination test (MAT) panel for the diagnosis of Leptospirosis in a low resource, hyper-endemic setting with varied microgeographic variation in reactivity.," has been formally accepted for publication in PLOS Neglected Tropical Diseases.

Best regards,

Shaden Kamhawi

co-Editor-in-Chief

Paul Brindley

co-Editor-in-Chief
